

# Impact of long-range transport over the Atlantic Ocean on Saharan dust optical and microphysical properties

Cristian Velasco-Merino[1], David Mateos[1], Carlos Toledano[1]*, Joseph M. Prospero[2], Jack Molinie[3], Lovely Euphrasie-Clotilde[3], Ramiro González[1], Victoria E. Cachorro[1], Abel Calle[1], Angel M. de Frutos[1]

[1]Grupo de Óptica Atmosférica, Dpto. de Física Teórica Atómica y Óptica, Universidad de Valladolid, Spain
[2]Cooperative Institute for Marine and Atmospheric Studies, Rosentiel School of Marine and Atmospheric Science, University of Miami, Miami, Florida, USA
[3]Laboratory of Geosciences and Energy, Université des Antilles, Guadeloupe

*Correspondence to*: C. Toledano (toledano@goa.uva.es) Office B110, Grupo de Óptica Atmosférica, Dpto. de Física Teórica Atómica y Óptica, Fac. Ciencias – UVa, Paseo Belén 7, CP 47011, Valladolid.

**Abstract.** Mineral dust aerosol can be a major driver of aerosol climatology in regions distant from the sources. This study addresses the change of columnar aerosol properties when mineral dust arrives to the Caribbean Basin after transport from Africa over the Atlantic Ocean. We use data from NASA Aerosol Robotic Network (AERONET) sites in five Caribbean and two West African sites to characterize changes in aerosol properties: aerosol optical depth, size distribution, single scattering albedo, and refractive indexes. After obtaining local aerosol climatology in each area, the air mass connections between West Africa and Caribbean Basin have been investigated by means of air mass back trajectories. Over the period 1996-2014 we identify 3174 connection days, on average, 167 connection days per year. Among these, 1162 pairs of data present aerosol data in Caribbean sites with corresponding aerosol observations in Western Africa sites ~5-7 days before. Of these 1162 days, 484 meet the criteria to be characterized as mineral dust outbreaks. Based on these days we observe the following changes in aerosol-related properties in transiting the Atlantic: AOD decreases about 0.16 or -30%; the volume particle size distribution shape shows no changes; single scattering albedo, refractive indexes, and asymmetry factor remain unchanged; the difference in the effective radius in West African area with respect to Caribbean Basin is between 0 and +0.3 µm; and half of the analyzed cases present predominance of non-spherical particles in both areas.

## 1 Introduction

Africa is one of the world's largest dust sources, estimated to produce over half the global total (Huneeus et al., 2011). Much of this dust is transported to the west and deposited over the Atlantic Ocean but much us carried reaches northern South America (Prospero et al., 2014; Yu et al., 2014, 2015), the Caribbean Sea (Prospero and Lamb, 2003), and southern United States (Bozlaker et al., 2013). Therefore, mineral dust serves as a key player in the planetary system with positive effects as a prolific source of nutrients to oceanic and forest ecosystems and significant planetary cooling, but with negative effects


related to human health transporting bacteria, degrading air quality, and affecting climate/meteorological patterns, among others (e.g., Prospero et al., 2005).

The observation of the Africa-Atlantic Ocean-America connection has been addressed since the 1970s using meteorological information linked to in-situ and satellite aerosol data (e.g., Prospero et al. 1981; Jickells et al., 2005; Rodríguez et al., 2015;

García et al., 2017). Recently developed techniques such as MACC (Monitoring Atmospheric Composition and Climate; Chouza et al., 2016) or MOCAGE (Modélisation de la Chimie Atmosphérique Grande Echelle; Martet et al., 2009) models have been also used to monitor dust plumes.

Studies dealing with long-term (e.g., Prospero and Lamb, 2003; Prospero and Mayol-Bracero, 2013) and short-term (e.g., Reid et al., 2003; Colarco et al., 2003; Kaufman et al., 2005; Valle-Díaz et al., 2016) databases show a seasonal dependence

in the intensity of the long-range transport, being more intense during boreal summer in the subtropical zone. Within this framework, the SALTRACE program was developed in 2013 (Weinzierl et al., 2017). The core of the SALTRACE program was an atmospheric column closure experiment, held in June/July 2013, involving ground-based and airborne in-situ and remote sensing observations in African and Caribbean sites. Preliminary results have pointed out that mineral dust makes a significant contribution to aerosol load in the Caribbean. During this campaign, the removal rate of large super-micron

particles seems to be slower than expected (Weinzierl et al., 2017) as previously reported by Maring et al. (2003).

Within SALTRACE, a unique Lagrangian in-situ study was carried out, wherein a dusty air mass was sampled over the Cape Verde Islands and again five days later over Barbados separated by a distance of more than 4,000 km. Our study is inspired in that experiment, although our approach is to use columnar aerosol data collected from CIMEL sun-photometers of the AERosol RObotic NETwork (AERONET) in the two areas. The amount of AERONET data is as large as 19 years of

ground-based measurements in various sites. Therefore, the aim of this work is to investigate the changes in dust optical and microphysical properties due to the long-range transport, by comparing AERONET observations in West Africa and the Caribbean Basin under a climatological perspective (period 1996-2014).

Section 2 presents the database and methodology used to match daily-averaged AERONET data at both sides of the Atlantic Ocean using air mass backwards trajectories. Section 3 describes, separately, the long-term aerosol climatology in West

Africa and the Caribbean Basin and the comparison between AOD and Dust concentration in the Caribbean Basin. Section 4 presents the monthly variability of aerosol size parameters. Section 5 presents the air mass Caribbean-African connection climatology, as well as the change in aerosol optical depth, size distribution, and single scattering albedo, among others after the long-range transport. Finally the conclusions are presented in Section 6.

## 2 Database and Methodology

**2.1 AERONET measurements and sites**

The main database for this study includes the daily mean values of columnar aerosol data measured by CIMEL CE-318 Sun photometer in the AERONET framework (Holben et al., 1998). The direct sun algorithm provides a database that contains



instantaneous values (every 15 minutes) of spectral AOD at 7 wavelengths in the range 340nm – 1020nm (in some cases also 1640nm) and the associated Ångström Exponent (AE) derived in the range 440-870 nm. We only use AOD and AE quality assured level 2.0 data (version 2) which ensures the reliability of the measurements and includes cloud-free screening. The AOD values presented in this study are only referred to the 440 nm wavelength; hence, from now on AOD indicates the

aerosol optical depth at 440nm.

Figure 1 and Table 1 present the five AERONET sites in the Caribbean (CAR) zone and the two sites in the African (AF) zone chosen in this study.

The CIMEL instrument hourly measures sky radiances both in almucantar and principal plane geometries at certain wavelengths in the range 340-1640 nm (number of wavelengths depending on the instrument model). The sky radiances,

together with the AOD, are used to derive optical and microphysical properties of the aerosol using inversion procedures (Dubovik et al., 2006). The inversion-derived parameters used in this study are: Volume Particle Size Distribution (VPSD) and the volume concentration for the fine, coarse and total size distribution ($VC_F$, $VC_C$, $VC_T$, respectively), Sphericity fraction (SF), Total Effective Radius ($ER_T$), Real Refractive Index (REFR), Imaginary Refractive Index (REFI), Single Scattering Albedo (SSA), and Asymmetry Parameter (g).

The Level 2 inversion requirement rejects AOD values less than 0.4 which dramatically reduces the amount of data. In order to increase the number of measurements in the data set we used level 1.5 inversion products but we applied an extra level of quality control to ensure the reliability of the inversion data:

-    VPSD and $ER_T$: same as AERONET level 2.0 criteria (solar zenith angle >50deg., number of symmetrical angles, and sky error between 5% and 8% depending on AOD, see

http://aeronet.gsfc.nasa.gov/new_web/Documents/AERONETcriteria_final1_excerpt.pdf), but without threshold with respect to AOD.

-    SSA, g, REFR, REFI and SF: same as AERONET level 2.0 criteria but with AOD ≥ 0.20 (see Dubovik et al., 2006; Mallet et al., 2013; Mateos et al., 2014) instead of >0.4.

The use of level 1.5-filtered data with an extra quality control has been previously used by other authors (e.g., Burgos et al.,

2016). All the available inversion products are daily averaged in this study.

Global climatology in West Africa and Caribbean Basin of key aerosol properties are addressed in Section 3. For this global study, all daily available records (see Table 1) in the two West African and five Caribbean sites are averaged in the 1996-2014 period, resulting two databases with 5656 and 5099 days of data, respectively.

## 2.2 Linkage between Caribbean and West Africa by air mass back-trajectories

All the air mass connections between West Africa and Caribbean Basin are searched. This task requires the separate analysis for each Caribbean site. For instance, the methodology used in this study for Barbados site is the following:

-    **Step1**: We calculated three-dimensional 10-day back-trajectories with 1-hour time resolution using the Hybrid Single Particle Lagrangian Integrated Trajectory Model (HYSPLIT) version 4.0 (Stein et al., 2015). The





geographical coordinates used as the start point are those presented in Table 1. As mineral dust can be transported to altitude levels higher than the boundary layer, most notably in the Saharan Air Layer (Carlson and Prospero, 1972; Yu et al., 2014; Groß et al., 2016), back-trajectories were calculated at 750, 2500 and 4500 m (a.g.l.) using the model vertical velocity in the calculations. The meteorological database used as input for HYSPLIT is the NCAR/NCEP

global Reanalysis (Kalnay et al., 1996). The evaluation is performed for each day in the period 1996-2014 at 16:00 UTC (around local noon).

-    **Step2**:  If any of these passes through a 3º×3º box centered on the West African sites - we consider that a link has been established between Caribbean and African sites. The transit time is typically about 5 – 7 days. In some cases an air mass back-trajectory passes through both Capo_Verde and Dakar boxes. In case the mean of the aerosol

observations will be used.

-    **Step3**: Once the dates in which the air mass is measured in Barbados and in Capo_Verde and/or Dakar sites are established, the associated aerosol data is assessed. In the case of Barbados site, the corresponding daily means of all AERONET products are used. However, the African site analysis requires a special procedure. The visual inspection of many of the cases shows that the aerosol records for the specific date obtained in the trajectory  analysis  often fell

on a day without data (e.g., due to cloudy or rainy conditions) or did not capture the central days of the dust event. To resolve this problem, we introduced a ±1 day adjustment to the back-trajectory estimated date at Capo_Verde and Dakar. This is in line with previous studies which state that desert dust episodes in West African sites usually last for several days (e.g., Knippertz and Stuut, 2014). For the Barbados-African cases with air masses overflowing only one of the sites, Capo_Verde or Dakar, the corresponding date-adjusted aerosol data are used.

These three steps described are applied to all Caribbean sites. The geographical coordinates used in this procedure are those presented in Table 1. Therefore, the outcomes of this methodology are five databases (one per site) in which each Caribbean site is connected with the African database. For each day between 1996 and 2014 we determine which Caribbean sites present an air mass connection with West African area.

We identify a desert dust event on the basis of the following criteria: AOD ≥ 0.2 and AE ≤ 0.6 (e.g., Dubovik et al., 2002;

Guirado et al., 2014). In the evaluated global Caribbean-African connected database, the two areas are analyzed separately. Hence, there are two different inventories that contain all days meeting the criteria for mineral dust in the Caribbean Basin and in West Africa. In the global database there are, therefore, three different cases which are extensively analyzed in Section 5:

    a)   $D_{AF} + D_{CAR}$: Desert dust conditions occur in both Caribbean and African areas

b)   $D_{AF} + NoD_{CAR}$: Dust condition in West Africa but non-dust condition in the Caribbean

    c)   NoD: Non dust conditions

We found a total of 3174 days showing connection between African and Caribbean areas. Just 1162 out of 3174 days present AERONET data (level 2.0) of AOD and AE. Of these 1162 days, 484 meet the criteria to be characterized as mineral dust outbreaks. For the analysis of microphysical and radiative properties, the number of matching inversion data is however



much smaller (just 71 cases). The frequent cloudiness in the Caribbean basin is the main difficulty for the inversion of sky radiances (see Table 1).

## 3 Seasonal variability of aerosol load at the West African and Caribbean sites

In this section the analysis of the monthly-seasonal cycle of the main aerosol properties in the two areas is presented
separately – that is there is no requirement that trajectories connect the African and Caribbean sites. This section summarizes overall multi-year statistics (from 1996 to 2014) of AOD, AE and VPSD quantities.

Figures 2 and 3 present the multiannual AOD and AE monthly averages of 19 years in African and Caribbean regions, respectively. There is a significant difference between the "cold" (October-April) and "warm" (May-September) seasons in both areas. We have held here the usual cold-warm distinction in northern hemisphere areas, but there is not that difference
in temperature seasonal cycle.

In the West African winter season there is a weak variation of AOD and AE (see Figure 2) with values around 0.3 and 0.3-0.5, respectively. In contrast in the summer season there is an increase of AOD together with a decrease of AE until June
when the maximum of AOD 0.6 coincides with a minimum of AE 0.15. This peak was also observed by, e.g., Tegen et al. (2013). After June there is a progressive decrease of AOD values and increase in AE until once again reaching winter levels. Overall, the large AOD and low AE values suggest dominant effect of coarse particles in dust episodes.  The ranges of AOD between 0.3 and 0.6 and AE below 0.5 were also obtained by previous studies in West African sites (e.g., Dubovik et al., 2002; Horowitz et al., 2017). Leon et al. (2009) reported that M'Bour site (western Senegal, 70km away from Dakar) is
under the major influence of desert dust and tropical biomass burning aerosol emissions throughout the year.

The annual cycle of AOD in the Caribbean (see Figure 3) shows an almost perfect bell shape with the maximum in June (AOD = 0.3) and the smallest values in December-January. In contrast, the AE displays a larger variability along the year: an increase from January to April when AE = 0.7 (the annual maximum), followed by a steady sharp decline until the absolute
minimum in June-July with AE= 0.26. Levels remain constant around 0.55 from September to December. This feature can be understood as the fingerprint of the occurrence of mineral dust transport from Africa. The AE variability evidenced by the standard deviations is large driven by the changes in aerosol mixture that occur in this area (e.g., Reid et al., 2003). Clean maritime conditions, associated to background values, have low AOD and low AE (Smirnov et al., 2000), but they can be modified by mineral dust outbreaks from African deserts and biomass burning episodes. In particular, the large change in AE
in the "cold" season is largely due to the advection of pollutant aerosols from higher latitudes (e.g, Savoie et al., 2002; Zamora et al. 2013).





Previous work (e.g., Smirnov et al., 2000) has shown that high concentrations of dust at the surface are correlated with high column optical depths measured by a collocated AERONET instrument. Yu et al. (2014) has shown that CALIPSO (Cloud-Aerosol Lidar and Infrared Pathfinder Satellite Observations) dust concentrations over Barbados track the surface-based measurements of dust. As a first step we show the relationship between the long-term measurements of dust concentrations

at the surface with the columnar aerosol load manifested in AOD. We use the surface dust concentration measured at Ragged Point, Barbados, site (e.g., Prospero and Lamb, 2003; Prospero and Mayol, 2013) between 1996 and 2011. Daily surface dust concentrations are obtained with high volume filter samplers from measured aluminum concentrations assuming an Al content of 8% in soil dust (e.g., Prospero, 1999) or from the weights of filter samples ashed at 500º C after extracting soluble components with water (e.g., Huneeus et al., 2011 and references therein).

Surface dust concentration seasonal cycle (Figure 3) presents a significant increase between low values during October-April season up to larger concentrations in May-September season.   Overall, the shapes of the annual cycles of AOD and surface dust concentration at Barbados are similar and the same seasonal pattern in both variables is observed.

Monthly mean AOD is highly correlated with surface dust concentrations as shown in Figure 4 which is based on 3700 pairs

of daily data, a total of 192 monthly mean, and 12 inter-annual monthly mean values. The seasonal distribution of the surface-columnar values seems to follow a linear increasing pattern from winter (bluish colors) to summer (reddish colors). The slope of the fit between dust concentration and AOD is about 115 µg m$^{-3}$ per unit of AOD for the two cases analyzed here: using 192 monthly means (Figure 4a) and 12 inter-annual monthly means (Figure 4b). The Barbados dust concentration vs AOD shows very good correlation when the same months are averaged along the years. However, the monthly agreement

between these two variables displays a certain degree of dispersion with a moderate correlation coefficient.  The negative intercepts in the fits are most likely attributable to the effects of sea salt aerosol on AOD.

## 4 Monthly variability of aerosol size parameters

Figure 5 displays the average monthly cycle of the volume particle size distribution (VPSD) for the African and Caribbean areas, respectively. These figures are based on 3346 and 2165 daily mean values of the AERONET inversion products

selected as described above.

The coarse mode predominance can be observed throughout the year in both areas. The seasonal cycle shape of VPSD in African area (Figure 5a) does not show any significant change through the year with peak concentration of the coarse mode about 2.24 µm. The magnitude of VPSD in the coarse mode ranges between maximum values in June with 0.32 µm$^3$/µm$^2$

and minima in November-December with 0.05 µm$^3$/µm$^2$. The fine mode plays an almost negligible role throughout the year. This seasonal pattern was already reported by previous studies in the African area (e.g., Dubovik et al., 2002; Eck et al.,



2010; Guirado et al., 2014). These studies showed the domination of large particles (radius beyond 0.6 μm) with VPSD peaks for coarse mode at 2 μm, which are independent of the aerosol load.

The conditions observed in the Caribbean area (Figure 5b) show a change in the size of the aerosol particles along the year. With respect to the coarse mode, in the "warm" season (May-Sep) the maximum concentration peaks about 2.24 μm, linked to mineral dust, and the shape of the size distribution is the same as it is found in the West African sites (Fig. 5a). The concentration values are lower (as it is AOD) but the shape is identical. In the "cold" season (Oct-Apr), however, the coarse mode maximum is achieved at larger radius (about 3.85 μm) highlighting the predominance of large sea salt particles. This effect has been observed in other coastal locations affected by dust outbreaks (e.g., Prats et al., 2011). In addition, the maximum volume concentration of the coarse mode exhibits maximum values in June with 0.12 $\mu m^3/\mu m^2$ and minimum ones in November-December with 0.02 $\mu m^3/\mu m^2$. The fine mode also plays an almost negligible role throughout the year in the Caribbean area. The whole curve of VPSD in the "cold" season is in line with previous studies carried out in Oceanic environments (e.g., Dubovik et al., 2002) with low particle volume concentrations peaking at very large radius (> 3 μm).

## 5 Air Mass Connections between African and Caribbean areas

In this section, we select only those days when the back trajectories connect the African area with the Caribbean as explained in Section 2.2. In this way, the changes in the aerosol properties observed in the Caribbean area are established by comparison with those observed some days earlier over the West African sites.

### 5.1 Air-mass African and Caribbean connections: all data and dusty data

Following the methodology described in the Steps 1 and 2 of Section 2.2, we obtained a list of all days with air mass Caribbean-West Africa connections. Figure 6 shows the seasonal cycle of the total number of days with this connection in the period 1996-2014. Overall, almost half of the days each year (~167 days per year) display Caribbean-African connection with a total of 3174 cases in 19 years.

The total number of air mass connections between both areas exhibits small values from January until the absolute minimum in April with only 2 days per year. This minimum of connections could be linked with the local minimum of aerosol load observed in the seasonal cycle in the West African area (see Figure 2).

With the beginning of the "warm" season (May-September) the number of connections shows a notable increase achieving its maximum in July with almost all the days in the Caribbean Basin being connected with West African area. From October to December there is a progressive decline from 20 to 10 connection days per year.

Unfortunately, columnar aerosol data are not available for each one of the 3174 connection days. A total of 1162 out of 3174 days (36% of the total) are present in AERONET data of level 2.0 (see Section 2.2). Furthermore, just 484 cases (15% of the total) meet the dusty criteria in both areas (case 'a)' described in Section 2.2). This is due to the limitation in the data



coverage and to the strict criterion used to unambiguously identify desert dust events. It should be noted that this procedure underestimates the actual number of desert dust events observed by ground-based measurements. For instance, during June-July-August in the Caribbean Basin there is essentially continuous dust (e.g., Prospero et al., 2015), which is corroborated in this study with an average of 23-28 connection days in these months.

**5.2 Scatterplot AE-AOD in the African and Caribbean areas**

The following step of this study focuses on the comparison of the aerosol properties observed in the Caribbean area and the values of the same properties which were observed days before over the African sites. In this way, the impact of the long range transport can be quantified. Figure 7 presents the scatterplot AE-AOD for those days with aerosol data and air mass connection between the two areas. When the criteria for identifying mineral dust (AOD ≥ 0.2 and AE ≤ 0.6) are applied to

the global Caribbean-African connected database, four different cases are identified (see Section 2.2).

In the African zone (Fig. 7a), as expected because of the proximity to the Saharan Desert, the influence of the mineral dust aerosol properties is predominant: 86% of the available data have AOD ≥ 0.2 and AE ≤ 0.6, (1000 out of 1162 days). We identified the occurrence of 498 dusty days in the Caribbean area, about 42% of the entire global Caribbean-African connected database, but just 484 days (in the period 1996-2014) are shown to be dusty days in the Caribbean area with

aerosol origin in the Saharan desert ($D_{AF}$ + $D_{CAR}$ case in Figure 8). Intense dusty days (AOD larger than 0.5) occur in West African and Caribbean in 430 and 67 cases, respectively. This difference is attributable to aerosol load loss during transport between the two regions. Out of the total of 1162 cases 516 meet the dusty criteria only in the African database ($D_{AF}$ + $NoD_{CAR}$ case in figure 8). This decrease in dust loads is also observed in the change of the AE and AOD using ground-based and satellite measurements (see, e.g., Yu et al., 2014). For instance, there are 102 cases in the Caribbean database where AE

> 0.6 whereas the same air mass yielded AE < 0.6 in the African zone days before. In addition, there are possible misidentifications of dusty days with AOD and AE values close to the required thresholds (156 and 59 days in the interval of 0.15 < AOD < 0.20 in the Caribbean and African zones, respectively). This misidentification can also occur in the "NoD" case, when non-dusty conditions occur. We considered lowering the threshold in AOD to lower values so as to include more cases but this would reduce confidence about the actual presence of dust.

**5.3 Differences between AOD in the African zone and the Caribbean area**

Once the connections have been detected, changes in AOD due to the long-range transport over the Atlantic Ocean are studied depending on the AE values. The results are shown in Figure 8. The $AOD_{CAR}$ has been plotted as a function of AOD in the African zone ($AOD_{AF}$).


The scatter plot in Fig. 8 shows that there is, as expected, a decrease in AOD in air masses transiting the Atlantic. The mean decrease of AOD between Africa and the Caribbean is 0.16, a decrease of about 30% with respect to the values at Dakar and





Cabo Verde. The decreases are related to dust concentration. They are greatest for cases where AOD values are large (AOD > 0.8) at the African sites, with decreases up to 70%. In the interval 0.5 < AOD < 0.8 the decreases are in the range about 28-45% while those in the interval 0.2 < AOD < 0.5 are in the range 11-27%. These percentages based on long-term and ground-based data are a first approach to assess the estimations done by studies dealing with satellite data. For instance, Yu

et al. (2015) have reported, in a latitudinal belt of 10ºS-30ºN, a loss of 70% (±45–70%) between dust amount (in Tg a$^{-1}$) leaving West Africa (at 15°W) and the end of the Caribbean area (at 75ºW). Hence, our estimations are in line with these findings considering that the measured variable and studied area are not identical.

The cases with low AE values are those cases with a wider variability in AOD difference. In addition, there are some cases in which AOD is larger in the Caribbean. This effect could be explained by different reasons: inhomogeneity of the dust

layer, addition of other aerosol particles in the Caribbean area (such as biomass burning), temporal gaps in the instantaneous measurements that cause non-representative daily averages, cloud contamination not detected by the cloud-screening algorithm, etc.

**5.4 Aerosol microphysical properties for African-Caribbean Sea connections**

In this section we address the key aerosol microphysical properties for those cases where we demonstrated a back-trajectory

connection between the African and Caribbean sites. Figure 9 shows the VPSD for the two cases analyzed before: dust conditions in both areas ($D_{AF} + D_{CAR}$) and only in Africa ($D_{AF} + NoD_{CAR}$). The shape of the VPSD curve for this collection of data in the African side is the same as described in detail in Section 4. The fine mode volume fraction (obtained as the ratio $VC_F/VC_T$) is on average as low as 0.077.

The VPSD shape in the Caribbean for the $D_{AF} + D_{CAR}$ case is the same as in the African area: average coarse mode volume

concentration of 0.22 µm$^3$/µm$^2$, maximum peak about 2.24 µm, minor role of fine mode (average fine mode volume concentration of 0.02 µm$^3$/µm$^2$ peaking at 0.086 µm, fine mode volume fraction of 0.09). The coarse mode volume concentration decreases from 0.34 µm$^3$/µm$^2$ in West Africa to 0.22 µm$^3$/µm$^2$ in the Caribbean Sea. Hence, there is a loss about 35% in the coarse mode concentration between both areas caused by the long-range transport over the Atlantic Ocean.

To quantify the change in the aerosol size caused by the long-range transport, Figure 10 shows the histogram of the differences between the effective radius of the total size distribution ($ER_T$) in the two areas. Most of the differences in the $ER_T$ values are positive, thus indicating the size distribution in West Africa generally having larger particles than in the Caribbean Sea. The differences in the effective radii are mostly confined (about 70% of the cases) between no change and a decrease in the effective radius about 0.3 µm between both areas. The maximum of occurrence is found for a decrease about

0.2 µm. Negative differences (15% of the cases), meaning larger particle size in the Caribbean area, could be attributed to the presence of other aerosol layer (e.g., sea salt) in the atmospheric column.





The third microphysical property studied in this study is the fraction of spherical particles found in the inversion process (for details, see Dubovik at al., 2006). The results are presented in Figure 11. This figure shows that in the African zone there is large predominance of non-spherical particles. About 80% and 53% of the values in the African and Caribbean zones correspond with basically complete non-sphericity (SF < 0.05), respectively. Overall, in 32 out of 71 cases (45%) there is no change between both areas and the shape of the particles is predominantly non-spherical. Cases with sphericity fraction below 0.05 in African sites display a wide variability in the observed fraction in the Caribbean sites, even achieving values of 0.7. This increase could be explained by the mixture of mineral dust with other (spherical) aerosol particles, such as maritime aerosols. Note that we use column observations; therefore the mixture in this case does not mean that dust and other aerosol particles are necessarily together. They can be separated in different atmospheric layers (e.g., elevated dust layer and boundary layer, Groß et al., 2015). The measurements at Barbados collect dust in the boundary layer (Prospero, 1999; Smirnov et al., 2000; Yu et al., 2014), so that dust simultaneously occur with sea-salt or other aerosol layers (e.g., Ansmann et al., 2009; Toledano et al., 2011; Groß et al., 2011, ; 2016). However the highest concentrations of dust are found in the elevated SAL (Yu et al., 2014) where there are no significant concentrations of sea salt aerosol (Savoie et al., 2002).

## 5.5 Aerosol radiative properties for African-Caribbean Sea connections

Figure 12 shows the main radiative properties (SSA, g, REFR, and REFI) in order to compare whether the optical properties of the mineral aerosol as detected in the African zone, remain the same after the long-range transport or exhibit any significant difference over the Caribbean. Taking into account the uncertainty estimates and the observed variability, the absorption power (SSA in Figure 12a and REFI in Figure 12c) is the same in both areas. The SSA at Dakar and Cabo Verde increases with wavelength: from 0.94 at 440 nm up to 0.98 at 670, 870, and 1020 nm. These figures are in line with previous studies in this area (Dubovik et al., 2002; Eck et al., 2010; Kim et al., 2011; Toledano et al., 2011; Giles et al., 2012, among others) but also at various Spanish and Mediterranean sites during desert dust outbreaks (e.g., Meloni et al., 2006; Cachorro et al., 2010; Valenzuela et al., 2012; Burgos et al., 2016). The SSA in dust events in the Caribbean Basin is essentially identical to that at the African sites. The same is true for REFI: 0.0025 at 440 nm and 0.001-0.0015 in the interval 670-1020 nm. The values in the visible and near infrared ranges are slightly larger than those reported by Dubovik et al. (2002) which were about 0.0007; their value at 440 nm is very similar to the our value. There is no significant change in the REFR either, being the mean real part of the refractive index in the African and Caribbean database, about 1.45 ± 0.01. This value is slightly lower than the reported by Dubovik et al. (2002) which was about 1.48. Finally, the asymmetry factor g at 440 nm decreases slightly from 0.77 in African sites to 0.755 in the Caribbean. The g values reported in this study are slightly larger than the reported by previous studies for desert dust in Africa (e.g., Dubovik et al., 2002). Overall, intensive optical properties (both absorption and scattering quantities) do not change in spite of the long range transport.



## 6 Conclusions

We characterized the changes in aerosol properties during long-range aerosol transport between Africa and Caribbean Sea using AERONET sun photometer measurements over the period 1996-2014. We selected two AERONET sites in the eastern Atlantic – Dakar and Cabo Verde – to characterize the properties of the air masses emerging from West Africa and five

AERONET sites in the eastern Caribbean to make identical measurements on these same air masses about one week later. We obtained over 5000 values that qualified for our study. The seasonal cycles of AOD (at 440 nm wavelength) and AE in the two areas are similar between May and September with maximum AOD (0.6 and 0.3 in Africa and the Caribbean Basin, respectively) and minimum AE (about 0.2 in both areas) in June. The volume particle size distribution (VPSD) shows the dominant influence of coarse particles throughout the year in both areas, with maximum values during the summer months

when dust transport is at a maximum.

We used HYSPLIT air mass back-trajectories to identify days when air parcels passing over the Caribbean sites also overpassed one or both of the African sites. Overall, 3174 trajectory-linked days were identified in the 1996-2014 period of which 1162 pairs of data qualified for aerosol comparison analysis. For the cases in which dust is detected at the African and Caribbean sites (484 cases, 15% of the total connection days), the AOD shows a mean decrease of about 0.16 AOD-units or -

30% after transit. The largest decreases were seen for the larger AOD values which imply larger losses during transport.

When aerosol inversion products are analyzed, only 71 days satisfied the selection criteria. In comparing inversions products before and after transit of the Atlantic, we could find no substantial changes in the shape of the volume particle size distribution and the spectral dependence of absorbing and scattering power. The analysis of the total effective radius shows larger aerosol particles, differences between 0 and +0.3 µm in the 70% of the cases, in the West African area. Non-spherical

particles are the predominant shape in both areas. The change of these quantities can be explained because of the mixture of mineral dust with other aerosol layers and the early deposition of larger particles. Therefore, intensive radiative properties do not change significantly in spite of the long range transport.

### Acknowledgments

The authors gratefully acknowledge the NASA AERONET program for the very valuable data used in this study. We thank

Brent Holben (Barbados, La_Parguera), Didier Tanre (Capo_Verde, Dakar), and Olga Mayol-Bracero (Cape San Juan) for their effort in establishing and maintaining their sites. This work has received funding from the European Union's Horizon 2020 research and innovation programme under grant agreement Nº 654109 (ACTRIS-2). The authors are grateful to Spanish MINECO for the financial support of the IJCI-2014-19477 grant, PTA2014-09522-I grant, and POLARMOON project (ref. CTM2015-66742-R). We also thank 'Consejería de Educación' of 'Junta de Castilla y León' for supporting the

GOA-AIRE project (ref. VA100P17). Prospero received support from NSF AGS-0962256 and NASA NNX12AP45G.



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



**Table 1. Geographical coordinates, period of AERONET data and total number of level 2.0 AOD daily data from the different sites used for the global African and Caribbean databases. To highlight the number of available inversion products, the number of daily data of volume particle size distribution (VPSD) is presented.**

| Area | Site | Coordinates (ºN, ºE, m a.s.l.) | Time period | Number of AOD / VPSD daily data |
|------|------|-------------------------------|-------------|-------------------------------|
| AF | Capo_Verde (CV) | (16.71, -22.93, 60) | 1993-2014 | 4635/1910 |
| | Dakar (DK) | (14.38, -16.95, 0) | 1996-2014 | 3593/2466 |
| CAR | Barbados (BA) | (13.15, -59.62, 114) | 1996-2000 | 938/50 |
| | Barbados_SALTRACE (BA) | | 2013-2014 | 181/7 |
| | Ragged_Point (RG) | (13.15, -59.42, 40) | 2007-2014 | 1768/415 |
| | Guadeloup (GU) | (16.22, -61.53, 39) | 1997-2014 | 1949/441 |
| | La_Parguera (LP) | (17.97, -67.03, 12) | 2000-2014 | 3303/1467 |
| | Cape_San_Juan (SJ) | (18.38, -65.62, 15) | 2004-2014 | 1901/401 |





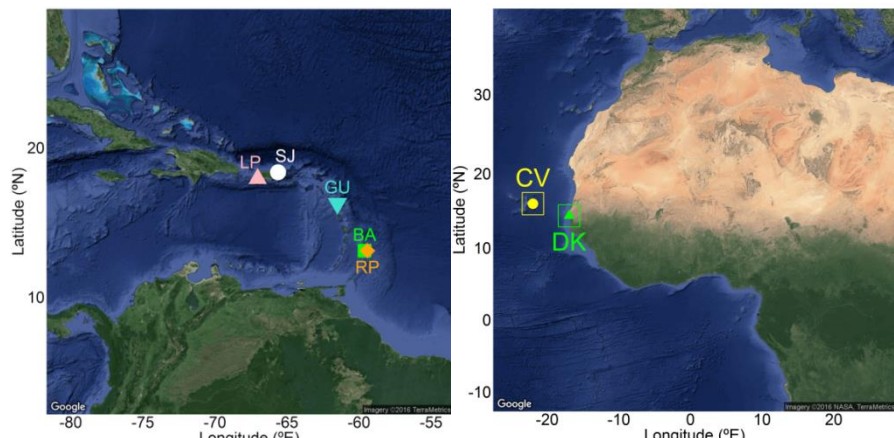

Figure 1: AERONET sites in the Caribbean Zone (left) and African zone (right). For acronyms see Table 1.




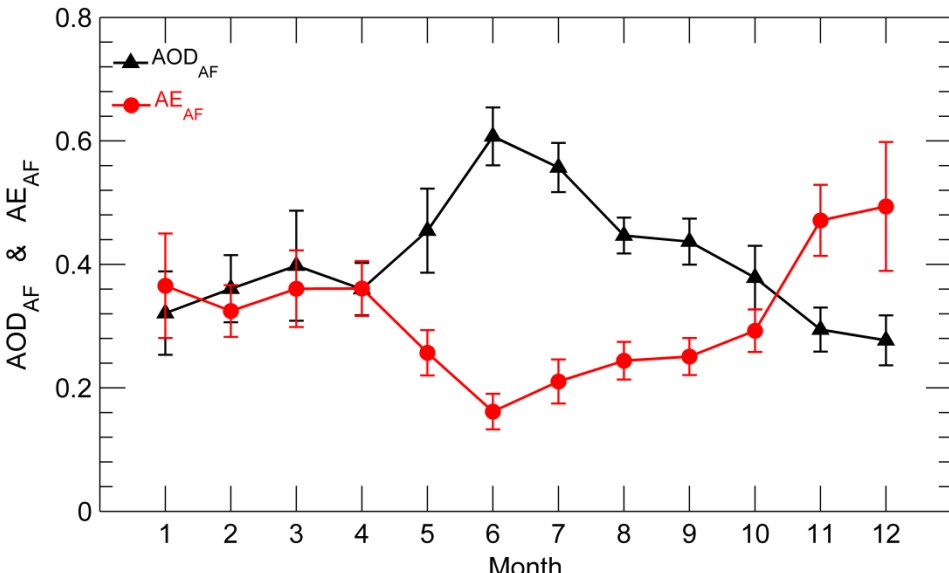

**Figure 2: Monthly average of AOD and AE from 1996 to 2014 in the African area. The number of daily means used in the multi-annual averages is 5656**.



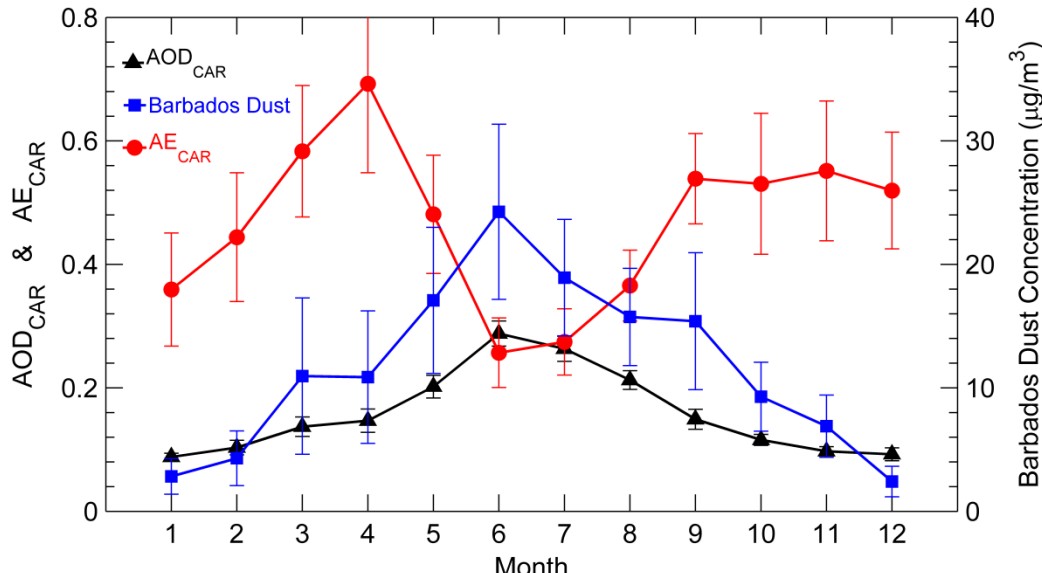

**Figure 3. Monthly average of AOD and AE in the Caribbean area from 1996 to 2014 and surface dust concentrations in Barbados in the period 1996-2011. The number of daily means used in the multi-annual averages is 5099 for AODCAR and AECAR and 5167 for surface dust concentration in Barbados.**





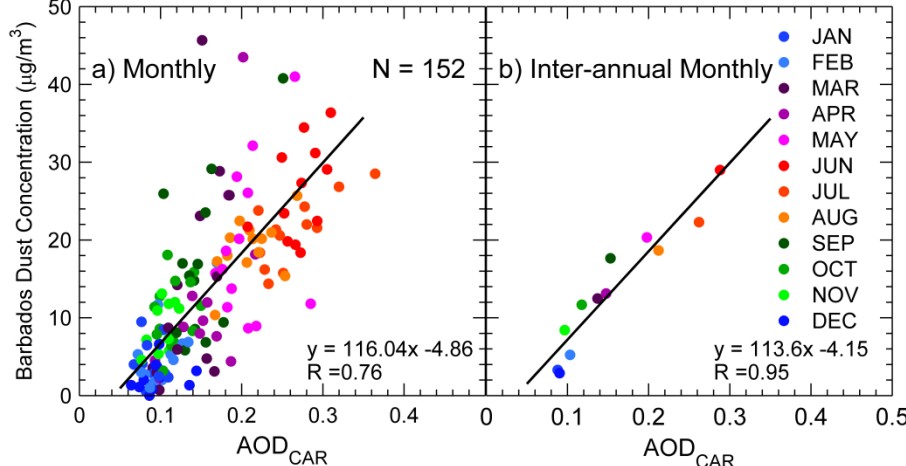

**Figure 4: Scatterplot of monthly values of surface dust concentration and AOD in the Caribbean Basin in the period 1996-2011. Solid line highlights the linear fit between both quantities.**





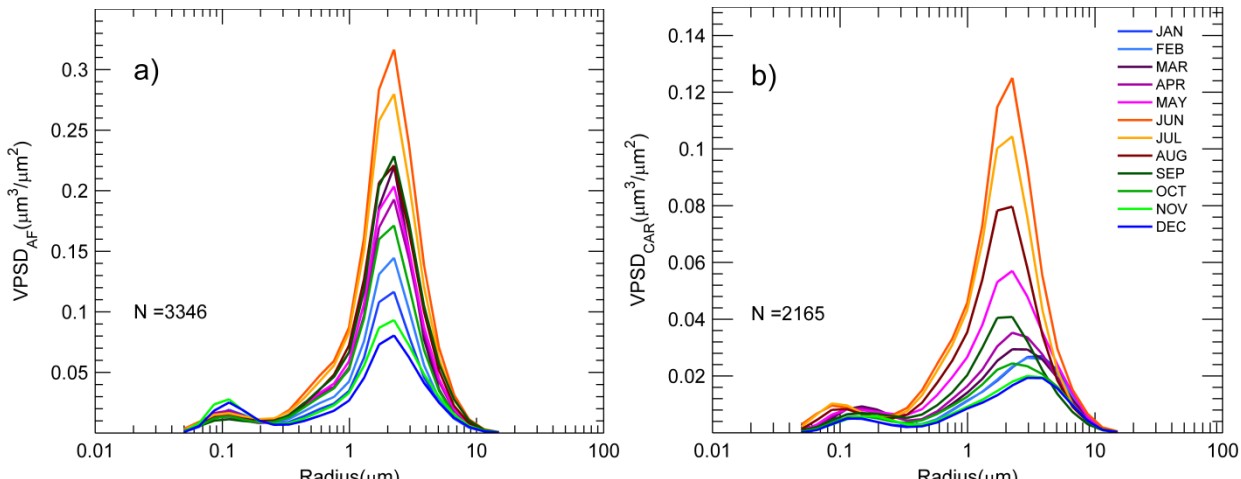

**Figure 5: Monthly averages of VPSD from 1996 to 2014 in West African (a) and Caribbean (b) areas.**





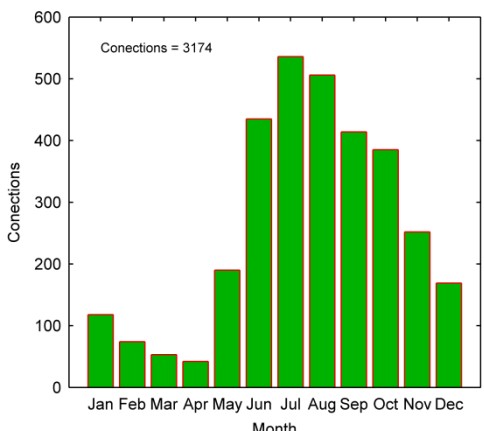

**Figure 6. Seasonal cycle of number of Africa-Caribbean Sea air mass connections.**





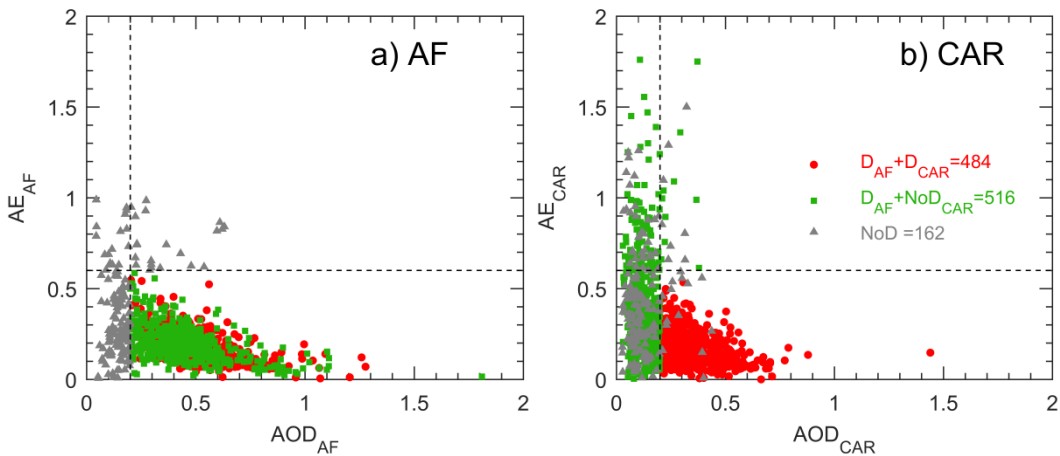

**Figure 7: AE-AOD scatterplot for all data days in a) West Africa and b) Caribbean Sea. Red circles are dust days in both Africa ($D_{AF}$) and Caribbean ($D_{CAR}$) databases, green squares are dust days in Africa which are not dusty in Caribbean area (NoD$_{CAR}$), , and grey triangles are non-dusty days (NoD) in both areas. Dashed lines indicate the criteria for identifying mineral dust (AOD ≥ 0.2 and AE ≤ 0.6).**





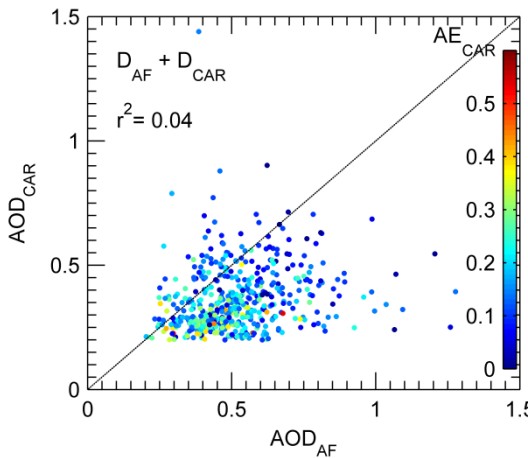

**Figure 8. Scatterplot of AOD in the Caribbean (AOD$_{CAR}$) versus AOD in the African zone (AOD$_{AF}$). The color bar indicates the AE in the Caribbean zone (AE$_{CAR}$) when dust is observed in both areas (DAF+DCAR case, see Section 2.2). Solid lines point out no change in AOD data between both areas.**





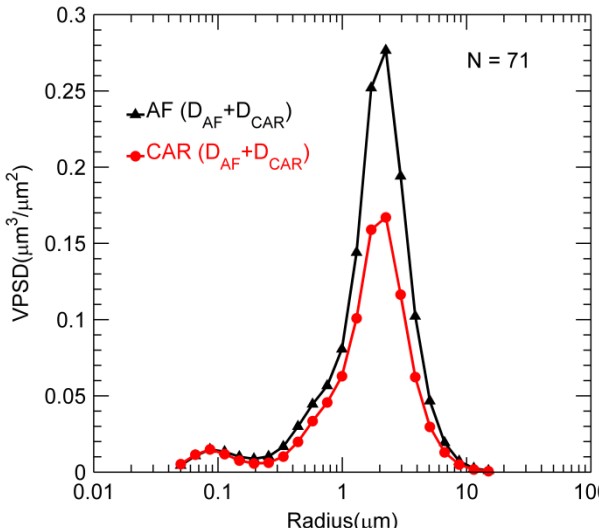

**Figure 9: VPSD in the African (AF) and the Caribbean (CAR) areas under desert dust conditions ($D_{AF}$ + $D_{CAR}$ case, see Section 2.2).**





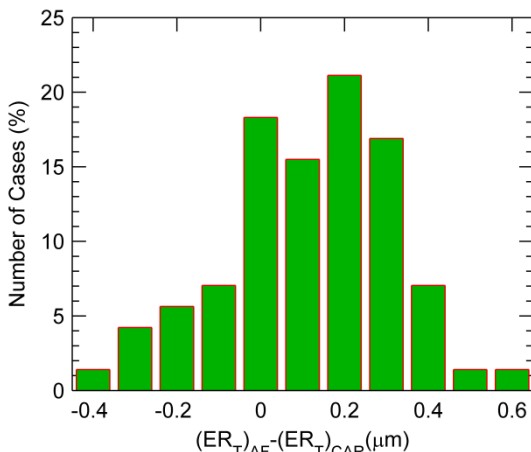

**Figure 10: Histogram of the differences between the total effective radii in the African zone and in the Caribbean area for those cases with connected desert dust conditions in both areas ($D_{AF} + D_{CAR}$ case).**



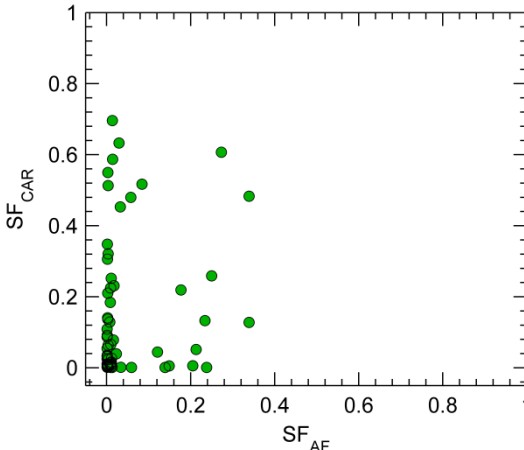

**Figure 11: Sphericity fraction values in the African zone and in the Caribbean area for those cases with desert dust in both areas.**





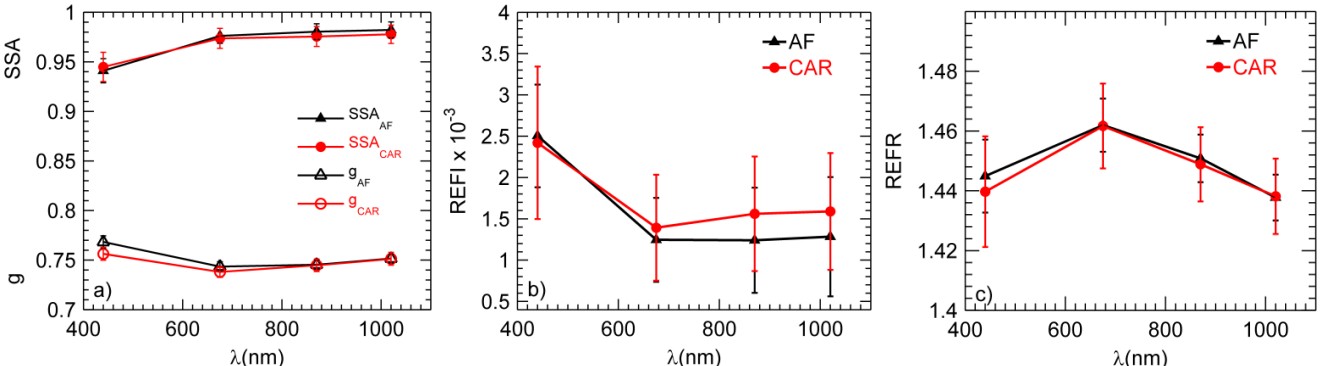

**Figure 12: a) Single scattering albedo (SSA) and asymmetry parameter (g), b) Imaginary part of the refractive index (REFI), c) Real part of the refractive index (REFR) for those cases with desert dust in both African zone (black lines) and Caribbean area (red lines).**