# Peer review of "Impact of long-range transport over the Atlantic Ocean on Saharan dust optical and microphysical properties"

_Atmospheric Chemistry and Physics, 2017_

## Referee Comment (RC1) · Anonymous Referee #1 · 16 Jan 2018

The discussion paper by Velasco-Merino et al investigates changes of the Saharan dust properties due to the trans-Atlantic transport on the basis of AERONET sun photometer data. The analysis is supplemented with trajectory simulations. Overall, this is a nice study covering almost 20 years of data. However many, mainly minor, improvements are necessary before publication in ACP is recommended.

General comments:

* The language needs to be improved. It would be really good if the paper is corrected by someone who speaks English as his/her first language. Though there is a "English copy-editing service" that the paper will go through before publication in ACP, the au-

thors should pay attention to correct spelling and grammar (according to the "General obligations for authors" of ACP).

\* "African zone", "African area", "West African area", "African sites" are used to describe the same thing. Same for the Caribbean. I suggest to always use "West African sites" and "Caribbean sites" to be clear.

\* Also the term for the trajectory-derived links varies and thus is not precise. I suggest to make this more clear, for example by avoiding "connection" and using only "links" throughout the paper.

\* In general, it should be mentioned in the legends (and where important in the text) what the uncertainties in the figures stand for and how they were derived.

\* In the discussion it is not mentioned that there is also some uncertainty in the AERONET products.

\* Furthermore, putting the results into context with existing literature could be done more extensively (for example using additional SALTRACE papers). Most evidently, the conclusion section lacks a discussion of the impact of the paper, which should be added.

Some specific comments:

\* Something like "based on AERONET data" should be added to the title to make clear that this study is based on AERONET data.

\* In the abstract it should be mentioned that AOD refers to 440nm.

\* Page 1 / Line 17ff.: I suggest "... we identify 3174 days on which the airmass over the Caribbean sites was linked to at least one of the two West African sites, which is on average 167 days per year. For 1162 of these days, AOD data is available for the Caribbean sites as well as for the corresponding West African sites about 5-7 days when the airmass overpassed these sites." (might be corrected if necessary)

[Figure]

* P1/L20: You need to include in the abstract how you characterize "mineral dust outbreaks" (high AOD, low AE on both side of the Atlantic?)

* P1/L23: I suggest to show "ER_CAR-ER_AF" not "ER_AF-ER_CAR" (also in the main text and Fig. 10) since you want to show the impact of long-range transport.

* P1/L23: The change of effective radius does not really fit to "the volume particle size distribution shape shows no change".

* P1/L24: Of which cases? Saharan dust should always be non-spherical.

* P1/L27: Suggestion: "... deposited into the Atlantic Ocean but significant amounts reach the northern ..."

* P2/L1: Please make more clear what you mean with "to human health transporting bacteria".

* P2/L3: Suggestion: "Observations of the dust transport from Africa over the Atlantic Ocean to the Americas were performed using meteorological ..."

* P2/L10: "in the subtropical zone" could be removed.

* P2/L12: "held" -> "performed"

* P2/L19: Suggestion: "The temporal coverage of the AERONET data is as large as 19 years from 1996 to 2014." (Remove "1996-2014" on line 22)

* P2/L25: "comparison" -> "correlation"

* P3/L3: "includes cloud-free screening" -> "is cloud-screened"

* P3/L4: "are only referred" -> "refer"

* P3/L4-5: The part starting with "hence" is redundant.

* P3/L9: Why write here 340nm when the lowest sky wavelength is 440nm? Or are there really sky measurements at 340nm? For consistence with P3/L1 (AOD range) I

would suggest to write 1020nm as the upper limit.

* P3/L8: Do you also use principle plane measurements? If not, this should be mentioned.

* P3/L26: Though "global" is not completely wrong here, I suggest to use a different term because now it also sounds like you have a world-wide climatology...

* P4/L3-4: "using the model vertical velocity": This sentence now sounds like the back-trajectories don't need horizontal velocity. Please rephrase.

* P4/L30: What about case "NoD_AF+D_CAR"?

* P4/L33: after "AE" you could insert: "The measurements at the other days are affected by clouds on one or both sides of the Atlantic" (or maybe for additional reasons?)

* P4/L34: "as mineral dust outbreaks". Is this when the criterion on line 24 is met on both sides of the Atlantic? Should be specified more clearly.

* P5/L9-10: It is not clear what this sentence means.

* P5/L13: Remove "West Africa" and add "at the West African sites" after "AE"

* P5/L20: "tropical biomass burning aerosol emissions throughout the year": I can not find this in Leon et al. (2009). Please check again.

* P5/L25: It is not clear what "This feature" refers to. Please rephrase.

* P6/L22: "aerosol size parameters" -> "size-related aerosol parameters" (or something else) - The term "size parameter" normally refers to the size of a particle relative to the wavelength.

* P6/L26: Please rephrase "The seasonal cycle shape of VPSD"

* P6/L30: "The fine mode plays an almost negligible role ": This is too general (also at some other places in the text). This is true for particle volume and mass. However for the optical properties, in particular at short wavelengths, the fine mode can not be

neglected. For things like particle number concentration (or for cloud condensation nuclei) the fine mode is dominant even during dust outbreaks.

* P7/L6: "but the shape is identical" is redundant.

* P7/L21: "3174 cases": It is not clear what happens in the case when for more Caribbean sites a trajectory link is found on one day. Is it counted as one day and the aerosol data of that day is averaged over the stations? Should be clarified.

* P7/L24: "only 2 days per year": I think "per month" would be better here (same on line 28)

* P7/L24: "could be linked" -> "coincides"

* P7/L30: Data present at both regions?

* P8/L12-15: So 16 days are "NoD_AF+D_CAR"?

* P8/L15+L18: Figure 7 not 8.

* P8/L23: Write "... lowering the AOD threshold so as to ..."

* P9/L1: The sentence "The decreases ..." could be removed in my view.

* P9/L16: "... and only in Africa (D_AF+NoD_CAR)" is not in Fig. 9

* P9/L30-31: Also uncertainties of the inversion can be a reason for negative differences.

* P10/L9: "together" -> "in the same layer"

* P10/L18: "absorption power" is proportional to AOD times (1-SSA). So this term is not correct here.

* P11/L3-4: "in the eastern Atlantic" -> "on the east side of the Atlantic"

* P11/L15: But the relative decrease is the same for smaller and for larger AOD values?

* P11/L19: Change of effective radius does not fit to line 17 ("no substantial changes in the shape of the volume particle size distribution")

* P11/L20: "The change of these quantities": It is unclear what this refers to because some before-mentioned quantities did not change.

* Table 1 legend: "period of AERONET data" -> "considered time period" ; "different sites" -> "different AERONET sites"

* Figure 6: Is it possible to divide the number of connection by number of months?

---

## Referee Comment (RC2) · Anonymous Referee #2 · 19 Jan 2018

This manuscript reports an analysis of nearly 20 year data from the AERONET stations in both West Africa and Caribbean Basin, assisted by HYSPLIT trajectory analysis. The study focuses on examining changes of dust properties (including loading, microphysical properties, and optical properties) along the trans-Atlantic transit. Result from this study complements what the SALTRACE campaign achieves and adds a useful piece to the characterization of trans-Atlantic dust transport. However, the presentation of paper should be improved either taking advantage of ACP's "English copy-editing service" or editing by a native English speaker.

A use of HYSPLIT trajectory analysis to establish connections between Caribbean

Basin and West Africa is interesting. Identified cases of the connection are useful to communities. I was wondering if authors can publish all the connection cases as supplementary material. I also would like to see some clarifications in 2.2. from the authors. For example, is the connection determined based on one of three altitudes (750m, 2500m, and 4500 m) or all three altitudes? For one day if all five sites in Caribbean Basin are connected to West Africa, does this count as one case or five cases? In Figure 6, does "3174 connections" mean "3174 days of connection"? It seems that 3174 has been interpreted as 3174 days in text.
* * *

---

## Referee Comment (RC3) · Anonymous Referee #3 · 22 Jan 2018

The study by Velasco-Merino et al. describes properties of dust that is transported from the Saharan region to the Caribbean Basin, based on almost 20 years of AERONET data. Trajectory calculations are done to identify mineral dust outbreaks. Although some improvements are necessary, it can be a valuable contribution on dust properties in this region. Particularly, the use of HYSPLIT for the selection of events is interesting, yet the current implementation may not be applicable.

General comments

To determine a link between the Caribbean and African sites, 10-days backwards particle trajectories were calculated starting from 3 heights at each site using HYSPLIT.

[Figure]

Details of these simulations are lacking, but it reads as if single trajectories were used. Given the uncertainty of these calculations, it is a rough assumption that there is a link (or not) if only a single trajectory passes one of the African sites. How many trajectories were calculated per day? Do you find the same number of links if you use multiple trajectories? (And consequently, does it change results of the analysis on particle properties?) Can you add statistics of how often the links were seen at all 5 Caribbean sites simultaneously?

The properties of dust are studied mostly based on column observations. Observations were thus influenced by the presence of other aerosol (layers). In case HYSPLIT trajectories at multiple heights indicate a link with African sites, other influences are of course not excluded but a better agreement might be visible in the data. Do you obtain similar results if you analyse such events specifically?

The conclusions that the volume particle size distribution shape shows no changes and the effective radius decreases appear contradicting.

Does the transport duration influence the observed changes?

Although the structure of the manuscript is clear, the text is at times hard to follow. There are many mistakes and naming is not consistent throughout the manuscript. I strongly recommend to carefully revise the manuscript (especially the conclusions), preferably with help of a native speaker.

Specific comments

Introduction: Since the aim of the study is to investigate changes in dust optical and microphysical properties during long-range transport you could add some words on these properties. What properties do you mean, why are they relevant, what is known already? The relevance of your findings on these topics should also be discussed more in the results/conclusions sections.

P4, Line 7 "if any of these passes". Do you mean any of these trajectories?

P4, Line 9-10 "In case .. will be used". Unclear

P4, Line 14 "often", can you quantify this?

P4, line 16 +/- 1 day adjustment; unclear

P4, Line 32 3174 days; Are these single days or summed from all 5 data bases? (For example, does 1 day with a link at each site count as 5?)

P5, line 10; unclear what you mean

P5, line 21; remove "almost perfect"

P7, line 5, shape of size distribution is the same; is this to be expected from previous studies? Please discuss.

P7, line 30, 15%; The percentage relative to the cases with data (1162) may be more relevant to report.

P8, line 10, "four"; should this be 3?

Section 5.2; It could be helpful to add a table with these numbers (number dusty days Sahara, number dusty days Caribbean etc.)

P9, line 30; It is clear that negative differences are influenced by other aerosols. Also positive differences may be influenced by other aerosols. The actual difference for mineral dust could thus be larger, this should be discussed.

P10, line 1; The inversion process needs more explanation than just a reference.

P11, line 15; You suggest that on days with larger AOD, losses during transport are larger. It could be interesting to look at particle properties on these days; e.g. are shifts in size distribution also larger on these days?

Figure 4, caption; please add an explanation of "Inter-annual Monthly".

---

## Author Comment (AC1) · 28 Mar 2018

The discussion paper by Velasco-Merino et al investigates changes of the Saharan dust properties due to the trans-Atlantic transport on the basis of AERONET sun photometer data. The analysis is supplemented with trajectory simulations. Overall, this is a nice study covering almost 20 years of data. However many, mainly minor, improvements are necessary before publication in ACP is recommended.

General comments:

* The language needs to be improved. It would be really good if the paper is corrected by someone who speaks English as his/her first language. Though there is a "English copy-editing service" that the paper will go through before publication in ACP, the authors should pay attention to correct spelling and grammar (according to the "General obligations for authors" of ACP).

**We have revised the English grammar and corrected errors in the text.

* "African zone", "African area", "West African area", "African sites" are used to describe the same thing. Same for the Caribbean. I suggest to always use "West African sites" and "Caribbean sites" to be clear.

**To homogenize the names throughout the manuscript, we followed the reviewers' suggestion.

* Also the term for the trajectory-derived links varies and thus is not precise. I suggest to make this more clear, for example by avoiding "connection" and using only "links" throughout the paper.

**It was changed.

* In general, it should be mentioned in the legends (and where important in the text) what the uncertainties in the figures stand for and how they were derived.

**According to the reviewers' suggestion, we have added the uncertainties of the inverstion products used in the text in the Section "2.1 AERONET measurements and sites".

* In the discussion it is not mentioned that there is also some uncertainty in the AERONET products.

As we mentioned in our previous comment, we have added the uncertainties of the AERONET inversion products in the methodology section.

* Furthermore, putting the results into context with existing literature could be done more extensively (for example using additional SALTRACE papers). Most evidently, the conclusion section lacks a discussion of the impact of the paper, which should be added.

**Following the reviewers' comments, we have added results into context with previous literature and, in particular, with SALTRACE campaign. For instance, in "Section 5.4

Aerosol microphysical properties for African-Caribbean Sea connections" we have added discussion and links about SALTRACE experiments and our results.

Some specific comments:

* Something like "based on AERONET data" should be added to the title to make clear that this study is based on AERONET data.

* In the abstract it should be mentioned that AOD refers to 440nm.

* Page 1 / Line 17ff.: I suggest "... we identify 3174 days on which the airmass over the Caribbean sites was linked to at least one of the two West African sites, which is on average 167 days per year. For 1162 of these days, AOD data is available for the Caribbean sites as well as for the corresponding West African sites about 5-7 days when the airmass overpassed these sites." (might be corrected if necessary)

* P1/L20: You need to include in the abstract how you characterize "mineral dust outbreaks"

(high AOD, low AE on both side of the Atlantic?)

**We have re-written all these sentences accordingly

* P1/L23: I suggest to show "ER_CAR-ER_AF" not "ER_AF-ER_CAR" (also in the main text and Fig. 10) since you want to show the impact of long-range transport.

**This change only means a change in the sign of the values, but it is interesting as you suggest in order to show the impact of the transport. We have changed the plot and the text in this sense. In the new version, most of the changes are negative and between 0 and -0.3 µm (P1/L29 & P10/L17).

* P1/L23: The change of effective radius does not really fit to "the volume particle size distribution shape shows no change".

**Yes, it was a bad sentence. We meant that some parameters like coarse mode effective and modal radii do not change, just the coarse mode concentration, but this really means a change in the fine/coarse fraction and the total effective radius, which decreases. We have changed the sentence accordingly (P1/L29 & P10/L17).

* P1/L24: Of which cases? Saharan dust should always be non-spherical.

**We have corrected the sentence. We mean that in some cases there are other aerosols present, and the overall (column effective) fraction of spherical particles is larger (P1/L29).

* P1/L27: Suggestion: "... deposited into the Atlantic Ocean but significant amounts reach the northern ..."

**We have re-written this section.

* P2/L1: Please make more clear what you mean with "to human health transporting bacteria".

**We have removed this sentence.

* P2/L3: Suggestion: "Observations of the dust transport from Africa over the Atlantic Ocean to the Americas were performed using meteorological ..."

**Done (P2/L11).

* P2/L10: "in the subtropical zone" could be removed.

* P2/L12: "held" -> "performed"

* P2/L19: Suggestion: "The temporal coverage of the AERONET data is as large as 19 years from 1996 to 2014." (Remove "1996-2014" on line 22)

* P2/L25: "comparison" -> "correlation"

* P3/L3: "includes cloud-free screening" -> "is cloud-screened"

* P3/L4: "are only referred" -> "refer"

* P3/L4-5: The part starting with "hence" is redundant.

**\*\*These 7 minor comments were corrected.**

\* P3/L9: Why write here 340nm when the lowest sky wavelength is 440nm? Or are there really sky measurements at 340nm? For consistence with P3/L1 (AOD range) I would suggest to write 1020nm as the upper limit.

\*\*The reviewer is right, and this is corrected in the new version (P3/L20).

\* P3/L8: Do you also use principle plane measurements? If not, this should be mentioned.

\*\*We have clarified this point, only almucantar inversions are used in the manuscript (P3/L19).

\* P3/L26: Though "global" is not completely wrong here, I suggest to use a different term because now it also sounds like you have a world-wide climatology...

\*\*We have decided to remove "global" in order to avoid misunderstandings.

\* P4/L3-4: "using the model vertical velocity": This sentence now sounds like the backtrajectories don't need horizontal velocity. Please rephrase.

\*\*"Model vertical velocity" is one of the options that Hysplit uses. The vertical motion can be solved with isobaric, isentropic trajectories, or using the vertical velocity provided by the model. This is the one we use, as it is commonly accepted to be more realistic. The sentence has been changed to make this point clearer (P4/L30 "step 1").

\* P4/L30: What about case "NoD_AF+D_CAR"?

\*\*Conditions with desert dust conditions in the Caribbean sites and non-desert in the African sites are less than 1% and they are included in NoD category. As we cannot ensure the presence of dust in African sites, we think that the option of 'NoD' category is the most suitable one for these cases.

* P4/L33: after "AE" you could insert: "The measurements at the other days are affected by clouds on one or both sides of the Atlantic" (or maybe for additional reasons?)

**Done (P6/L6).

* P4/L34: "as mineral dust outbreaks". Is this when the criterion on line 24 is met on both sides of the Atlantic? Should be specified more clearly.

**We have clarified this point (P6/L4).

* P5/L9-10: It is not clear what this sentence means.

**We think that this point is clearer in the new version (P5/L28).

* P5/L13: Remove "West Africa" and add "at the West African sites" after "AE"

**Done

* P5/L20: "tropical biomass burning aerosol emissions throughout the year": I can not find this in Leon et al. (2009). Please check again.

**At the end of the first column in page 2 of Leon et al. (2009) you can find the following sentence: "The M'Bour site is located in an area where the contribution of biomass burning aerosol is expected to be significant, in addition to mineral dust influence …". We have added more references about this topic (P6/L5).

* P5/L25: It is not clear what "This feature" refers to. Please rephrase.

**We mean "These features in the AOD and AE seasonality" (P6/L14).

* P6/L22: "aerosol size parameters" -> "size-related aerosol parameters" (or something else) - The term "size parameter" normally refers to the size of a particle relative to the wavelength.

**Done (P7/L8) .

\* P6/L26: Please rephrase "The seasonal cycle shape of VPSD"

\*\*Done. We have re-written this paragraph (P7/section 4).

\* P6/L30: "The fine mode plays an almost negligible role ": This is too general (also at some other places in the text). This is true for particle volume and mass. However for the optical properties, in particular at short wavelengths, the fine mode can not be neglected. For things like particle number concentration (or for cloud condensation nuclei) the fine mode is dominant even during dust outbreaks.

\*\*That's absolutely true. We have added "In terms of volume concentration…" in order to focus the discussion as desired. The role of fine mode particles is of course nonnegligible even at low concentrations. We have re-written this paragraph (P7/section 4).

\* P7/L6: "but the shape is identical" is redundant.

\*\*That part of the sentence was removed.

\* P7/L21: "3174 cases": It is not clear what happens in the case when for more Caribbean sites a trajectory link is found on one day. Is it counted as one day and the aerosol data of that day is averaged over the stations? Should be clarified.

\*\*We have clarified this point in the methodology. The following sentence has been added after the 3 steps are described: "If more than one site presents this kind of connection, we use the mean properties as representative." (Section 2.2 P5/L5)

\* P7/L24: "only 2 days per year": I think "per month" would be better here (same on line 28)

\*\*Changed (P8/L12)

* P7/L24: "could be linked" -> "coincides"

**Changed (P8/L12)

* P7/L30: Data present at both regions?

**We mentioned "see Section 2.2" to clarify this point when the whole database is explained. The 1162 days mean that there are measurements in the Caribbean sites with the corresponding data in African sites days before. (P8/L17)

* P8/L12-15: So 16 days are "NoD_AF+D_CAR"?

**There are 14 cases in the category that the reviewer mentioned. Thus, they are less than 1% of the database. The "NoD_AF+D_CAR" cases could be linked to possible bad characterization of dust plumes over West African sites. So, we decided to include these cases as NoD category, as was mentioned before, because the presence of African dust cannot be ensured.

* P8/L15+L18: Figure 7 not 8.

**Changed

* P8/L23: Write "... lowering the AOD threshold so as to ..."

**Done

* P9/L1: The sentence "The decreases ..." could be removed in my view.

**Removed

* P9/L16: "... and only in Africa (D_AF+NoD_CAR)" is not in Fig. 9

**Changed

* P9/L30-31: Also uncertainties of the inversion can be a reason for negative differences.

**Added. We have re-written section 5.4

* P10/L9: "together" -> "in the same layer"

**Changed. We have re-written section 5.4

\* P10/L18: "absorption power" is proportional to AOD times (1-SSA). So this term is not correct here.

**Removed

\* P11/L3-4: "in the eastern Atlantic" -> "on the east side of the Atlantic"

**Changed

\* P11/L15: But the relative decrease is the same for smaller and for larger AOD values?

**In that sentence we mentioned the mean decrease which is about 30%. We have explained this in detail in section 5.3: "The mean decrease of AOD between West African and the Caribbean sites is 0.16, a decrease of about 30% with respect to the values at Dakar and Cabo Verde. The decreases are related to dust concentration. They are greatest for cases where AOD values are large (AOD > 0.8) at the West African sites, with decreases up to 70%. In the interval 0.5 < AOD < 0.8 the decreases are in the range about 28-45% while those in the interval 0.2 < AOD < 0.5 are in the range 11-27%.". We have re-written section 6.

\* P11/L19: Change of effective radius does not fit to line 17 ("no substantial changes in the shape of the volume particle size distribution")

**Correct. This topic was clarified in a previous comment and has been corrected in the text. We have re-written section 6.

\* P11/L20: "The change of these quantities": It is unclear what this refers to because some before-mentioned quantities did not change.

**We want to describe the changes of effective radius and sphericity. We have rephrased the sentence. We have re-written section 6.

\* Table 1 legend: "period of AERONET data" -> "considered time period" ; "different sites" -> "different AERONET sites"

**Changed

\* Figure 6: Is it possible to divide the number of connection by number of months?

**We followed the reviewer suggestion, and the figure was changed accordingly. We provide in the reviewed version of the manuscript the number of air mass links per month, which is more illustrative here.

[revised manuscript text omitted]

---

## Author Comment (AC2) · 28 Mar 2018

This manuscript reports an analysis of nearly 20 year data from the AERONET stations in bothWest Africa and Caribbean Basin, assisted by HYSPLIT trajectory analysis. The study focuses on examining changes of dust properties (including loading, microphysical properties, and optical properties) along the trans-Atlantic transit. Result from this study complements what the SALTRACE campaign achieves and adds a useful piece to the characterization of trans-Atlantic dust transport. However, the presentation of paper should be improved either taking advantage of ACP's "English copy-editing service" or editing by a native English speaker.

**We have carefully revised the text again.

A use of HYSPLIT trajectory analysis to establish connections between Caribbean Basin and West Africa is interesting. Identified cases of the connection are useful to communities. I was wondering if authors can publish all the connection cases as supplementary material.

**We do not think this is necessary, mainly because anyone can easily reproduce the trajectory analysis with HYSPLIT and prepare the database adapted to their specific purposes or sites. Of course, we are interested in further collaborations and we can offer our inventory for future studies.

I also would like to see some clarifications in 2.2. from the authors. For example, is the connection determined based on one of three altitudes (750m, 2500m, and 4500 m) or all three altitudes?

**The three altitudes are investigated. If just one of them presents connection, the link is proved. We have added in section 2.2 and step 2: "If any of these links (at one, two or three heights) through a 3⁰x3⁰…"

For one day if all five sites in Caribbean Basin are connected to West Africa, does this count as one case or five cases?
**To clarify this topic we have added in section 2.2: "If more than one site presents this kind of connection (for each day), we use the mean properties as representative." So it would count as one.

In Figure 6, does "3174 connections" mean "3174 days of connection"? It seems that 3174 has been interpreted as 3174 days in text.

**It is the same. We are dealing with daily averages throughout the manuscript. When a connection is found, it is based on the daily averages.

[revised manuscript text omitted]

---

## Author Comment (AC3) · 28 Mar 2018

The study by Velasco-Merino et al. describes properties of dust that is transported from the Saharan region to the Caribbean Basin, based on almost 20 years of AERONET data. Trajectory calculations are done to identify mineral dust outbreaks. Although some improvements are necessary, it can be a valuable contribution on dust properties in this region. Particularly, the use of HYSPLIT for the selection of events is interesting, yet the current implementation may not be applicable.

General comments

To determine a link between the Caribbean and African sites, 10-days backwards particle trajectories were calculated starting from 3 heights at each site using HYSPLIT. Details of these simulations are lacking, but it reads as if single trajectories were used. Given the uncertainty of these calculations, it is a rough assumption that there is a link (or not) if only a single trajectory passes one of the African sites.

**We have added all the details used as input in HYSPLIT model. We are aware about the uncertainties of air mass back-trajectories. In the first step of this research, we have manually inspected most of the cases mentioned in the paper. In spite of the uncertainty of air mass trajectories, they are a good indicator of possible linkages between Africa and Caribbean Basin. As we could only use two sites in West Africa (it is a pity, but the number of available AERONET sites in West African area is not too large), we need a tool to check if the same air mass is measured in the African and in the Caribbean sites based on this two starting points. We could have used ensembles of trajectories, but we tried a simpler approach, by including 3ºx3º lat x lon box around each site to take in account variability or uncertainties of the simulated trajectories. We are not looking for exact points but just a reasonable link between Caribbean and African area.

How many trajectories were calculated per day? Do you find the same number of links if you use multiple trajectories? (And consequently, does it change results of the analysis on particle properties?)

**Three backwards trajectories (at three different heights) were evaluated ending at 16:00 UT (noon local time in the Caribbean Basin). We tested the possibility of adding more times in the simulation of air mass trajectories and we found that the results did not change significantly (nor the conclusions of the paper concerning aerosol property change after transport). Bear in mind that we open a window of ±1day in the African aerosol database that also considers uncertainties of the air masses. We decided to use only one time Hysplit calculations per day for simplifying the entire procedure.

Can you add statistics of how often the links were seen at all 5 Caribbean sites simultaneously?

**The problem at this point is the temporal coverage of the Caribbean data. The cloudy tropical systems make it difficult to have "continuous" records of columnar aerosol properties. This is the reason why we decided to use the "global Caribbean database", using the 5 individual datasets. Of course there are many events that can be seen in the entire Caribbean Basin, but there are also temporal delays between Southern/Northern and Eastern/Western areas that make difficult the joint analysis of the same event. As we mentioned in Section 2, we work with the 5 sites independently. After the first analysis, we decided to put all the connections together. This is the only way to increase the statistics about West Africa-Caribbean connections.

The properties of dust are studied mostly based on column observations. Observations were thus influenced by the presence of other aerosol (layers). In case HYSPLIT trajectories at multiple heights indicate a link with African sites, other influences are of course not excluded but a better agreement might be visible in the data. Do you obtain similar results if you analyse such events specifically?

**We have only analyzed microphysical and optical properties for the $D_{AF}$ + $D_{CAR}$ cases, so the presence of other aerosol types is minimized (but, of course, it is not excluded). It is true that other layers can also modify the columnar values of the aerosol properties even for predominant dust episodes. For instance, the sphericity fraction values beyond 0.05 reported in Figure 11 are an indicator of this fact. For future research, we are planning to study the vertical distribution of aerosols by LIDAR techniques (ground-based or satellite) and surface and columnar aerosol data in a closure experiment during the identified dust episodes in the Caribbean Basin. We need the vertical structure of each episode to well understand the mixture conditions observed from columnar values.

The conclusions that the volume particle size distribution shape shows no changes and the effective radius decreases appear contradicting.

**Yes, it was a bad sentence. We meant that some parameters like coarse mode effective and modal radii do not change, just the coarse mode concentration, but this really means a change in the fine/coarse fraction and the total effective radius, which decreases. Therefore there is a change in the shape of the size distribution. We have modified the sentence accordingly. We have re-written this part, section 5.4.

Does the transport duration influence the observed changes? Although the structure of the manuscript is clear, the text is at times hard to follow. There are many mistakes and naming is not consistent throughout the manuscript. I strongly recommend to carefully revise the manuscript (especially the conclusions), preferably with help of a native speaker.

**We have not investigated whether the changes in aerosol properties are significantly different as a function of transport duration, but this would be hard to assess in view of the back-trajectory uncertainties mentioned in previous comments. Bear in mind that we used a time window of ±1day in the West African database. So it is difficult to assess the right

length of transport. For future studies, we can address this kind of evaluation using other techniques such as maps from satellite observations.

About the text, we have tried to solve the problem about naming and revise the grammar.

The new version of the manuscript has been reviewed by all the co-authors again.

Specific comments
Introduction: Since the aim of the study is to investigate changes in dust optical and microphysical properties during long-range transport you could add some words on these properties. What properties do you mean, why are they relevant, what is known already? The relevance of your findings on these topics should also be discussed more in the results/conclusions sections.

**Thank you. We have added changes in the conclusions according to the relevance of our findings. We have re-written this section.

P4, Line 7 "if any of these passes". Do you mean any of these trajectories?

**Yes, we have changed passes by links. (P4, Line 21)

P4, Line 9-10 "In case .. will be used". Unclear

**Rephrased in the new version. (P4, Line 23)

P4, Line 14 "often", can you quantify this?

** 6% in AOD/AE available data and 24% in inversion available products.

P4, line 16 +/- 1 day adjustment; unclear
**To minimize uncertainties in the date of air mass trajectories estimations over West African sites, we have introduced ±1 day adjustment to the back-trajectory estimated date at Capo_Verde and Dakar. So, the aerosol records (AOD, size distribution, etc.) for African sites are averaged between date_estimated -1day and date_estimated + 1day. This choice is based on the manual inspection of most of the events presented in the manuscript. In this way, we can increase the statistics without much loss of accuracy. We have added the following sentence in Step 3 of Section 2.2: "The aerosol values for each West African site are therefore averaged considering the three days (estimated date and ±1day)".

P4, Line 32 3174 days; Are these single days or summed from all 5 data bases? (For example, does 1 day with a link at each site count as 5?)
**We have added extra details about the methodology in section 2.2. If there are links in different Caribbean sites for the same day, we select as representative the average between the sites involved and it only counts as 1.

However, in most of the cases each day only presents data in one Caribbean site. The same dust episode can be identified in the Caribbean sites with up to several days of delay regarding the geographical position.

P5, line 10; unclear what you mean

**We think it is clearer in the new version: "We have held here the usual cold-warm season distinction in northern hemisphere areas, but there is not that difference in temperature seasonal cycle in the Caribbean Basin". Just to remember that the cold-warm classification is typical in mid-latitudes of Northern Hemisphere. (P5, Line 27)

P5, line 21; remove "almost perfect"

**Done.

P7, line 5, shape of size distribution is the same; is this to be expected from previous studies? Please discuss.

**We mean the fact that coarse mode effective radius for sea salt is larger than for Saharan dust. Therefore, as described in Prats et al., during dust outbreaks at the coastal site (in southern Spain), the coarse mode concentration increases but the coarse mode effective radius and peak radius decreases. This is also the case in the Caribbean sites when we compare the size distributions, in particular the coarse mode, during background (marine) conditions and during dusty conditions. The sentence has been changed to: "This difference in the coarse mode radii between marine aerosol and dust has been observed in other coastal locations affected by dust outbreaks (e.g., Prats et al., 2011)." We have re-written section 4.

P7, line 30, 15%; The percentage relative to the cases with data (1162) may be more relevant to report.

**As the entire discussion in the paragraph is done using the total number of data, we thought it is less confusing if all the percentages are referred to the same amount. As the numbers of all the cases are explicitly given, the reader can calculate other percentages if desired. (P8, Line 17)

P8, line 10, "four"; should this be 3?

**Yes, thank you. It was a typo.

Section 5.2; It could be helpful to add a table with these numbers (number dusty days Sahara, number dusty days Caribbean etc.)

**The number of cases in the three categories ($D_{AF}$+$D_{CAR}$, $D_{AF}$+$NoD_{CAR}$, NoD) is included in Figure 7. The number of inversion products is also mentioned in Figure 9. The number of data used in the climatological analysis is also given in Figures 3 to 5 and their legends. As the paper has a large number of figures, we prefer to avoid including more tables.

P9, line 30; It is clear that negative differences are influenced by other aerosols. Also positive differences may be influenced by other aerosols. The actual difference for mineral dust could thus be larger, this should be discussed.

** We have re-written section 5.4. Both the mixtures (in the column) and possibly retrieval uncertainties are discussed as possible effects that must be considered for a correct interpretation of the results. This is a limitation of the measurement technique and the retrieval.;

P10, line 1; The inversion process needs more explanation than just a reference.

**The following sentence has been added to the text: "In the retrieval, particles are modeled both as spheres and spheroids, and the inversion finds what fraction of spherical and non-spherical particles (defined by aspect ratios) better fits to the observations".
The portion of spheres is therefore in the range [0,1]. We have re-written section 5.4

P11, line 15; You suggest that on days with larger AOD, losses during transport are larger. It could be interesting to look at particle properties on these days; e.g. are shifts in size distribution also larger on these days?

Following your suggestion, we analyzed the size distribution for the three cases of AOD losses. For cases with AOD > 0.8, VPSD decreases up to 67% (AOD decreases up to 70%). In the interval 0.5 < AOD < 0.8, the VPSD decreases about 40% (AOD decreases in the range about 28-45%). While in the interval 0.2 < AOD < 0.5, VPSD decreases about 28% (AOD loss is in the range 11-27%). There is a correlation between the decreases of both quantities. We have added this sentence to the conclusions: "Similar decrease is observed in the volume concentration of the size distributions." We have re-written section 6.

[Figure]

Figure 4, caption; please add an explanation of "Inter-annual Monthly".

**We have calculated the 'Inter-annual monthly mean' as the average obtained using 
[revised manuscript text omitted]

---

## Author Response (AR2)

**Co-Editor Decision: Publish subject to minor revisions (review by editor)** (17 May 2018)
by Bernadett Weinzierl
Comments to the Author:
Overall, the paper clearly improved, but there are still many typos and smaller errors left. In particular the new parts of the manuscrip have numerous spelling, punctuation, and other small errors: e.g. P1L24, P1L25, P2L3 (terestrial + buget), P2L9, P2L28, P2L33, P3L15, P5L4 ('an data'), P5L6, P7L22, P8L19, P10L25, P11L4

We've corrected all the mentioned typos. A careful revision was carried out.

The authors should carefully work through the manuscript and improve the points mentioned above. In addition, they should also address the specific comments detailed below.

Specific comments:

P1L24-25: "Comparing the AOD properties" doesn't make much sense here.

We agree and the sentence has been removed. We rephrased the whole sentence: "…mineral dust outbreaks. We observe that AOD at 440nm decreases about 0.16 or -30% after transit. The volume particle…"

P3L6: "aerosol size parameter" should be replaced by "size-related aerosol parameters".

OK.

P3L7: "air mass Caribbean-African connection climatology" should be improved.

The new sentence states: "Section 5 presents the seasonality of Africa-Caribbean Sea air mass connections,…"

P3L17: "zone" should be replaced by "region" or "area" (also in Fig 1, 8, etc) as suggested by one of the reviewers in the initial reviews

We've used the same words throughout the entire manuscript in the new version.

P3L23: Please clarify here (and in Fig. 5+9) that the VPSD means dV(r)/dln(r).

We agree that the physical meaning of VPSD should be mentioned. We have added the explanation in text and figures.

P3L24: "Real Refractive Index" should be replaced by "real part of the refractive index", same for the imaginary part.

These two terms are changed accordingly in the text.

P4L4: "no loss of accuracy" is probably too strong and according to Tab. 2 of Mateos et al. 2014 the statement seems to be wrong as there is some loss of accuracy.

We agree, we have rephrased the sentence: "with minor loss of accuracy (Mateos et al., 2014)."

P4L5-6: here the abbreviations like REFR etc. could be used

Changed.

P4L8: "in the ... sites" seems a bit unusual to me. I would suggest to write "at the ... sites" instead. This occurs many times in the paper.

Changed.

P4L12: "for Barbados" does not fit well to step 1 and 2 because the description of the methodology is more general. So you should describe step 1 and 2 using the example of Barbabos.

We have mentioned "Barbados" in the places of step1 and step2 when the particular info for this site is used in the methodology. In this way, the method is easily applicable to other sites.

P5L14: it should be mentioned that case c) also includes NoD_AF+D_CAR even if it does not occur often. Or just write "c) NoD: No-dust conditions at the Caribbean sites".

We agree. More explanation is required here. The case c) was rephrased: " c) NoD: No-dust conditions or dust conditions at the Caribbean sites with not proven origin.". We think this sentence fits much better with all the cases in NoD category.

P5L15: "showing connection" -> "with connections"

Replaced.

P5L18: "however" is repeated.

Removed.

P5L27-28: This sentence is not unclear. Please clarify. Maybe the sentence also could be removed.

We tried to avoid possible misunderstandings with cold/warm seasons in the Caribbean latitudes, by analogy with the other boreal areas. Because we mentioned the months included in each category in the previous sentence, we have decided to remove this unclear sentence.

P6L15: "large" -> "largely"

Changed

P6L23: "As a first step we show" -> "In Fig. 3 we show"

Changed

P7L15: it is probably sufficient to say "at about 2 µm" given the inversion uncertainties and the resolution of the size grid. This occurs also at other places in the paper where it also should be corrected.

We agree, it makes more sense to say "at about 2 µm", given the bin resolution of the VPSD.

P7L24: "achieved at larger radius (about 3.85µm)" -> "shifted to larger radii (> 3µm)"

Changed

P8L5: The title of section 5.1 is not well chosen. I suggest something like "Overview over air mass connections".

You are right; your suggestion is more adequate. The title has been changed.

P8L18: You should add here that you mean level 2 of AOD and AE data.

We have Added: "AERONET AOD and AE data" in the text.

P9L9: Maybe "misidentification" is not the best word because they were identified correctly as AOD < 0.2. I suggest "probably there are days with presence of dust with AOD ..."

We agree, we have rephrased the sentence: "Probably there are days with presence of dust with AOD and AE values close…"

P9L10-11: I think this sentence could be removed.

We agree, we have removed it.

P9L12: "so as to lower values" could be removed.

Removed.

P9L14: The title of section 5.3 is not optimal. If "Scatterplot" is left in the title of Section 5.2, it could be rephrased as 5.3 "Scatterplot of connected AOD data"

We agree, it would be more coherent. The title has been changed.

P9L25: "between" -> "of the" ; "leaving" -> "between"
P9L31: "etc" is not good, "for example" might be better.
P10L3: "the VPSD" -> "the average VPSD"
P10L9: "is a loss about 35%" -> "is a loss of about 35%"
P10L23: I think "(defined by aspect ratios)" is not necessary here.
P10L31: "during SALTRACE" -> "during the SALTRACE"
P11L21: " the reported " -> "those reported"

The last 7 minor changes have been modified accordingly.

P11L28-29: "quite similar from event to event": Where is this shown in the paper?

You are right, that conclusion is not directly shown in the paper (it is based on previous experience working with dust data over Africa). So, we have decided to change first paragraph of the conclusions and remove the sentence: "Focusing first on the results observed from the two African sites, we find that the dust properties are quite similar to those previously reported for Saharan dust in field experiments and previous studies using AERONET data."

Fig 3 legend: "AODCAR and AECAR": CAR as subscript; also not correct at other places in the paper

We corrected the subscripts.

Fig 4 lenged: "of:" -> "of"

Changed.

Fig 6: "Monthly/Annual mean Links" might be improved to "Average number of links per month/year"

We agree and the text was modified accordingly.

Fig 6: The labels on the x-axis should be homogenized to the labels in Fig. 2+3

Ok, we changed the x-axis.

Fig 7 legend: ", , and grey triangles are non-dusty days ("NoD") in both areas": This doesn't make sense because there are grey triangles in the "dusty region" of subplot b

We have clarified this point. We have removed "in both areas" in the 'NoD' case. We have added "see text in section 2.2" in order to refer the description of the three cases a), b) and c) in that section.

Fig 8 legend: "bar" could be removed; "DAF+DCAR" subscript missing; you should add that the plot is for days where a link was found.

We have corrected the subscripts and the caption has been modified: "Figure 8. Scatterplot of AOD in the Caribbean ($AOD_{CAR}$) versus AOD in the African region ($AOD_{AF}$) when dust is observed in both areas ($D_{AF}+D_{CAR}$ case, see Section 2.2). The color indicates the AE in the Caribbean area ($AE_{CAR}$). Solid lines point out no change in AOD data between both areas."

Fig 12 legend: It should be mentioned that these are average values.

We have added "Average values of…"

In general the figure legends should be homogenized (e.g. write everywhere "DAF + DCAR case" where applicable , e.g. in Fig. 11 and 12

We have homogenized all the captions.

[revised manuscript text omitted]

---

## Author Response (AR3)

**Co-Editor Decision: Publish subject to technical corrections** (10 Jun 2018) by Bernadett Weinzierl

Comments to the Author:

1) In the abstract, please change "We observe that AOD at 440nm decreases about 0.16 or -30% after transit." to "We observe that the AOD at 440nm decreases by about 0.16 or 30% during transport."

It has been changed.

2) In the legend of Fig. 6 the first "total" should be removed.

Done

3) In Fig. 8 "r^2=0.04" is written. It is not explained what this means. Please clarify. (In Fig. 4 "R" is written which probably is the correlation coefficient.)

We have changed r^2 by R and we have added "R is the correlation coefficient" in both figures.